# Antifungal Potential of Canarian Plant Extracts against High-Risk Phytopathogens

**DOI:** 10.3390/plants11212988

**Published:** 2022-11-05

**Authors:** Carolina P. Reyes, Samuel Rodríguez Sabina, Rocío López-Cabeza, Cristina G. Montelongo, Cristina Giménez, Ignacio A. Jiménez, Raimundo Cabrera, Isabel L. Bazzochi

**Affiliations:** 1Instituto Universitario de Bio-Orgánica Antonio González, Departamento de Bioquímica, Microbiología, Biología Celular y Genética, Universidad de La Laguna, Avenida Astrofísico Francisco Sánchez 2, 38206 La Laguna, Tenerife, Spain; 2Departamento de Botánica, Ecología y Fisiología Vegetal, Facultad de Ciencias, Sección Biología, Universidad de La Laguna, Avenida Astrofísico Francisco Sánchez, 38206 La Laguna, Tenerife, Spain; 3Instituto Universitario de Bio-Orgánica Antonio González, Departamento de Química Orgánica, Avenida Astrofísico Francisco Sánchez 2, 38206 La Laguna, Tenerife, Spain

**Keywords:** plant extracts, natural fungicides, phytopathogenic fungi, pest control

## Abstract

Phytopathogens are responsible for great losses in agriculture. In particular, *Fusarium*, *Alternaria* and *Botrytis* are fungal diseases that affect crops worldwide. In the search for eco-friendly solutions to pest control, plants and their chemo-biodiversity are promising sources of biopesticides for integrated pest management. The aim of the present study is to report the evaluation of sixteen plant species from the Canary Islands Archipelago against the phytopathogenic fungi *Botrytis cinerea*, *Fusarium oxysporum,* and *Alternaria alternata.* The plants were selected on the basis of their traditional uses in medicine and/or pest control, as well as on scientific studies reporting their uses in crop protection. Their growth inhibition (% I), in an in vitro test-assay on mycelium, was used to identify six ethanolic plant extracts displaying activity (% I > 30% at 1 mg/mL) against at least one of the assayed fungi. The most effective plant extracts were further fractionated by liquid–liquid partition, using solvents of increasing polarity. This procedure led to an improvement of the bioactivity against the phytopathogens, even affecting the hexane fraction from *S. canariensis* and achieving an 83.93% of growth inhibition at 0.5 mg/mL on *B. cinerea*. These findings identified five plant-derived extracts as potential candidates for the future development of new biofungicides, which could be applied in integrated pest management.

## 1. Introduction

Though the use of synthetic pesticides has offered significant economic benefits by enhancing food production and preventing vector-borne diseases, evidence suggests that their use has adversely affected the health of human populations and the environment [1]. In fact, several synthetic pesticides have been withdrawn from the market due to environmental issues resulting from their intensive use in fields and a reduction in their effectiveness because of an increasing resistance to target organisms [2]. Therefore, there is growing interest in developing alternative strategies towards an integrated form of pest control [3]. In this regard, biopesticides [4], in particular plant-derived pesticides, represent promising candidates, since they have some benefits over synthetic pesticides, such as being less toxic to humans, less persistent in the environment, and possessing new modes of action and molecular chemistry, and they can be used in organic agriculture [5]. 

Among the various diseases that severely affect crops, fungal infections via ascomycetes are of great incidence worldwide and synthetic fungicides have been widely used to fight these phytopathogens [6]. However, some of these fungicides, such as chlorothalonil, fenamidone, propineb, quinoxyfen, thiram, propiconazole, and mancozeb, have been withdrawn and/or banned in the European Union in recent years [7]. Furthermore, many target pathogens have developed resistance to fungicides, thereby compromising pest control [8]. Moreover, mycotoxin contamination decreases product quality, has repercussions for human health, and causes significant economic losses [9]. Among the phytopathogenic fungi, *Botrytis cinerea*, *Fusarium oxysporum,* and *Alternaria alternata* are responsible for plant diseases in economically important crops. In fact, *B. cinerea* is an airborne plant pathogen that causes grey mold in more than 200 crops all over the world [10]. The sesquiterpenes botrydial and botcinin acid are the two major phytotoxins involved in the infection caused by this fungus [11]. This pathogen can infect the plant products from the seedling stage until the product’s ripening. Even the harvested product can spoil since this fungus can survive as sclerotia and dormant mycelia both in field and storage facilities under extreme conditions. Thus, this pathogen is considered high-risk and has been classified as the second most important plant pathogen according to Dean et al. [12]. Moreover, *F. oxysporum*, a soil-borne pathogen that causes Fusarium wilt in crops such as melons, tomatoes, cotton, and bananas, produces a range of mycotoxins, including enniatins, fusaric acid, and moniliformin [13], and is considered one of the 10 most important fungi from a scientific/economic point of view [12]. On the other hand, *A. alternata* is a fungus that can affect a broad range of crops (potato, apple, and tomato) worldwide and is responsible for several diseases during the postharvest shelf life [14]. The main mycotoxins found in rotten samples of crops infected by this fungus are altenuene, alternariol, and altertoxins [15]. In the Canary Islands, the gray mold caused by *B. cinerea* usually affects tomato crops grown in unheated greenhouses during the winter [16], and *F. oxysporum* affects banana crops, with an incidence ranging from 2 to 12% [17], and several species of Palmaceae, including the Canary palm (*Phoenix canariensis*) [18]. Furthermore, *A. alternata* strongly attacks potato and vine crops, which are relevant crops in the economy of the Canary Islands and various studies have pointed out the development of cross-resistance by this phytopathogen [19]. 

In the search for eco-friendly solutions for pest control and environmental management, the interest in plants and their chemo-biodiversity as a source of biopesticides for integrated pest management has increased over time [5,20]. In fact, previous studies have reported the bio-efficacy of several plant extracts against *B. cinerea* [10,21], *A. alternata* [21], and *F. oxysporum* [22]. The Canary Islands archipelago has a surprising ecological diversity due to their special environmental conditions, which have given rise to a rich and varied flora with numerous endemic species [23]. Thus, these plants could be potential sources of biofungicides and a viable alternative to agrochemicals. Despite the possible antifungal potential of Canarian herbal extracts, the studies addressing this issue are scarce. One such study is the one by Coşovenau et al. [21], which reported the fungicidal activity of extracts from *Artemisa thuscula* and *Argyranthemum frutescens* against relevant phytopathogens.

Reported herein is an evaluation of 16 ethanolic plant extracts from the Canary Islands against the high-risk phytopathogenic fungi *B. cinerea*, *F. oxysporum,* and *A. alternata*. The studied plants were selected based on ethnobotanical knowledge as well as scientific reports on their potential in pest control. Six extracts showing high fungicidal activity were further fractioned by liquid–liquid partition, using solvents of increasing polarity, hexanes, ethyl acetate, and water, and the enriched fractions were also assessed against the target phytopathogens. 

## 2. Results and Discussion

### 2.1. Selection of Plants under Study

Many of the phytochemicals reported from medicinal plants (alkaloids, phenylpropanoids, terpenoids, or flavonoids) have been demonstrated to be effective in crop protection; therefore, the study of medicinal plant extracts [24] is becoming a field of growing interest as part of integrated pest management. 

In the present work, sixteen plants from the Canary Islands were studied, and the details of their collection, including the ecological state, place of collection, geographic coordinates, and voucher specimens are summarized in Table 1. 

The plants were selected for study based on their traditional uses as medicinal plants and/or in pest control (Table 2) and on scientific studies reporting their use in crop protection. The previous reports of the plants under study are briefly discussed. 

*Ceropegia fusca* is a plant used in traditional medicine, and the antimicrobial and antifungal activity of the extracts from this plant against clinic fungi have been reported [25], although its use in pest management has not been explored. Two species belonging to the Asteraceae family, *Argyranthemum broussonetti* and *Artemisia thuscula*, have also been included in the present study, as the later has been previously reported as an insecticide in potato crops [26] and as a fungicide against various phytopathogens [21]. Several compounds (sesquiterpenes) isolated from *Gymnosporia cassinoides*, formerly known as *Maytenus canariensis*, have shown both antifeedant and insecticidal activities against larvae of *Spodoptera littoralis* [27,28]. However, the antifungal potential of this plant has not been previously studied. The *Cistus* genus contains about 21 species, and extracts from some species of this genus were reported to be effective against *Geotrichum citri-aurantii* [29] and *G. candidum* [30], fungi that cause sour rot in citrus crops worldwide. Therefore, *Cistus symphytifolius*, an endemic species to the Canary Islands, and whose antifungal potential has not been previously assessed, was also selected for the present study. 

To the best of our knowledge, there have been no previous studies on the use of *Lavandula canariensis* and *Salvia canariensis* in pest control, which are two endemic species belonging to the Lamiaceae family. Nevertheless, the presence of compounds such as diterpenes [31] and triterpenes [32] with known fungicidal properties makes their study of interest. The study of *Apollonias burbujana* and *Laurus novocanariensis* (Lauraceae family) as fungicides could be promising, since several studies report the essential oils from *L. novocanariensis* as insecticides [33] and fungicides [34]. *Ruta pinnata* and *Ruta chalepensis* are two species of the Rutaceae family. The essential oils of *R. chalepensis* have shown fungicide activity against fungi of *Fusarium* [35], *Alternaria* [36], and *Aspergillus* [37] genera. 

Several plant species belonging to the Solanaceae family have also been selected, including *Datura stramonium*, *Datura innoxia, Nicotiana glauca*, *Salpichroa origanifolia,* and *Withania aristata*. Various studies on *D. stramonium* and *D. innoxia* have revealed their potential as biopesticides against different pests [38], including fungi [39,40]. Moreover, the acetone extract of *N. glauca* has shown antifungal activity against phytopathogenic fungi [41]. *S. origanifolia* is characterized by a high content of withanolides, metabolites that act as feeding inhibitors of insect species such as *Tribolium casraneum* [42]. Species of the genus *Withania*, which includes *W. aristata,* are also a rich source of withanolides [43]. The high content of this type of metabolite in both *S. origanifolia* and *W. aristata* suggest these plants could be promising candidates for sources of natural fungicides against the selected phytopathogenic species.

### 2.2. Screening for Antifungal Bioactivity of Plants’ Ethanolic Extracts

In the first step, the ethanolic extracts of the plants were evaluated against three phytopathogenic fungi: *Botrytis cinerea*, *Fusarium oxysporum,* and *Alternaria alternata* (Figure 1, and Appendix A). The extracts were firstly screened at a concentration of 1 mg/mL, and those exhibiting an inhibition higher than 20% were tested at lower concentrations, namely, 0.5 and 0.1 mg/mL. Methylparaben [57], a mold inhibitor, was used as a positive control, showing a 100% inhibition of the three pathogens at the concentration of 1 mg/mL. 

The crude ethanolic extracts showing bioactivity, namely, % growth inhibition (% I) > 10%, at the concentration of 1 mg/mL against at least one of the tested fungi were *C. fusca*, *A. thuscula*, *C. symphytifolus*, *S. canariensis*, *L. novocanariensis*, *R. pinnata*, *R. chalepensis*, *D. innoxia,* and *W. aristata*. *C. fusca* showed moderate activity against *B. cinerea* (% I = 28.91 at 1 mg/mL), exhibiting one of the lowest mycelial growth inhibition percentages compared to the other active plant extracts against this fungus. The extract of *A. thuscula* was more active against *F. oxysporum* (% I = 41.21% at 1 mg/mL) than against the other tested phytopathogens (% I = 23.63 and 12.41% for *A. alternata* and *B.cinerea* at 1 mg/mL, respectively). Coşoveanu et al. [21] reported similar results in their studies on *A. thuscula* leaf extracts. The ethanolic extract from *C. symphytifolius* exhibited antifungal activity against the three phytopathogens tested, ranging from 35.27 to 39.08% at 1 mg/mL. Therefore, the extract from this plant could be used as a broad-spectrum fungicide. However, the effect of the extract at lower concentrations is dependent on the fungus. In the case of *A. alternata* and *F. oxysporum*, the percentage of inhibition decreased considerably, becoming inactive for *Fusarium* fungus. By contrast, this extract showed similar potency against *B. cinerea* at 0.5 mg/mL (% I = 37.98%), and slightly lower at 0.1 mg/mL (% I = 19.57%). Two of the main metabolites reported in the *Cistus* genus are terpenoids and phenylpropanoids [58,59], whose antifungal activity has been previously studied. Moreover, several labdane diterpenes have been reported from *C. symphytifolius* [60,61] that could contribute to the bioactivity of this plant. Regarding *S. canariensis*, it was active against all the fungi evaluated, exhibiting a % I higher than 40% for all the fungi at 1 mg/mL. The activity was dose-dependent, although even at the lowest assayed concentration (0.1 mg/mL) the extract was able to inhibit fungal mycelial growth. Previous phytochemical studies on this species report the isolation of sterols, diterpenes, sesquiterpenes, and triterpenes [61], and some of these components exhibit an antifungal potential against the phytopathogens tested in this work. In fact, phenolic diterpenes such as carnosol and carnosic acid are active against *B. cinerea* [62], and triterpenes, including ursolic and oleanolic acids, are effective against *A. alternata* [63]. Therefore, the combination of these fungicidal compounds could be the responsible for the observed bioactivity of *S. canariensis* against the three fungi. *L. novocanariensis* showed a low activity against *F. oxysporum* (% I of 16.60 at 1 mg/mL), whereas it was inactive against the other tested fungi (% I < 10 mg/mL). The *R. pinnata* extract was moderately effective against *B. cinerea* and *F. oxysporum* at 1 mg/mL (28.27% and 25.47%, respectively). In the case of the other Rutaceae species studied, *R. chalepensis* showed a high activity against *F. oxysporum* (% I of 44.09% at 1 mg/mL), and a low activity towards *A. alternata* (% I of 17.08 at 1 mg/mL). Nevertheless, as the concentration decreased to 0.5 mg/mL, the % I dropped dramatically to 9.31%. *D. innoxia* exhibited antifungal activity against the three fungi tested, being more active against *B. cinerea* (% I = 52.11%), followed by *A. alternata* (% I = 37.95%), and exhibiting the lowest activity for *F. oxysporum* (% I = 20.74%) at 1 mg/mL. The antifungal activity of some plants belonging to the Solanaceae family has been associated with their content of tropane alkaloids [40,64], which, according to several studies, are high in *D. innoxia,* which was collected in different parts of the world [38,40,65]. Therefore, the great effectiveness of the *D innoxia* ethanolic extract against the tested fungi could be attributed to its content of tropane alkaloids such as hystocyamine and scopolamine [65]. *W. aristata* was active against the three phytopathogens. In particular, this species was the most active one against *B. cinerea*, showing a 72.22% growth inhibition at 1 mg/mL. An abundant group of withanolides have been identified in this plant [61], among which withanolide A exhibited antifungal and insecticide properties. 

These results revealed that *S. canariensis* and *D. innoxia* ethanolic extracts were the most effective against *A. alternata*, showing a % I of 44.61% and 37.95%, respectively, at 0.5 mg/mL. The extract from *W. aristata* was the most remarkably active against *B. cinerea*, followed by those from *S. canariensis* and *D. innoxia.* Furthermore, those extracts from *S. canariensis*, *R. chalepensis,* and *A. thuscula* showed a higher % I against *F. oxysporum* at 1 mg/mL. Moreover, *R. chalepensis* was more pathogen-selective, affecting only *F. oxysporum*, whereas *S. canariensis* could be used as a broad-spectrum fungicide, exhibiting high potency against the three evaluated phytopathogens. Furthermore, the most sensitive fungus to the plant extracts seems to be *B. cinerea*, followed by *F. oxysporum*, whereas *A. alternata* turned out to be the least sensitive to the studied plants. Thus, seven plant extracts showed a degree of growth inhibition higher than 20% against *F. oxysporum* compared to six plant extracts against *B. cinerea*. However, the % I values against *B. cinerea* were higher (28.27–72.22% at 1 mg/mL). Moreover, five plant extracts showed a % I > 20% against *A. alternata,* with values ranging from 22.99 to 47.60% at 1 mg/mL.

### 2.3. Liquid–Liquid Partition of the Most Effective Extracts

Those extracts exhibiting higher % I (>30%) against at least one of the target fungi were submitted to a liquid–liquid partition into three fractions—a hexane fraction (F1), an ethyl acetate fraction (F2), and an aqueous fraction (F3)—which were tested on the phytopathogens (Figure 2, Figure 3 and Figure 4, and Appendix A). 

The fractions from *A. thuscula* were tested on *F. oxysporum*, the most sensitive fungus to the ethanolic extract of this plant (% I > 30%). However, the fractionation did not increase its fungicidal activity (41.21% at 1 mg/mL), suggesting a synergistic effect between the components in the ethanolic extract that disappears when fractioning. Regarding *C. symphytifolus,* the percentage of inhibition increased with the fractionation for the three essayed fungi. This fact was noticeable in the case of *B. cinerea*, where the % I of the fractions C-F1 (88.43 and 85.80% at 1 and 0.5 mg/mL, respectively) and C-F2 (84.72 and 80.54% at 1 and 0.5 mg/mL, respectively) were two-fold higher than the ethanolic extract at 1 (39.08%) and 0.5 (37.98%) mg/mL, and three-fold higher at the lowest assayed concentration, 0.1 mg/mL. Thus, both the hexane and ethyl acetate fractions showed high activity, suggesting that different polarity compounds could be responsible for the fungicidal activity of this plant. The hexane (S-F1) and ethlyl acetate (S-F2) fractions from *S. canariensis* inhibited the mycelial growth of the three phytopathogens. These results are consistent with the fact that the diterpene and triterpene components are responsible for the antifungal activity of this plant, which are hydrophobic and, therefore, soluble in organic solvents [62,63]. A similar situation was observed in the case of *R. chalepensis* against *F. oxysporum,* because the lower polarity of the organic phases, the higher the percentage of mycelial inhibition. These results are in agreement with previous studies, which reported the antifungal activity of the essential oils from *R. chalepensis* [35,37]. In contrast, in the case of *D. innoxia*, the fraction showing less or no antifungal activity was the hexane fraction (D-F1), and its effectiveness against phytopathogens was higher by increasing the polarity. These results are in agreement with previous studies that attributed the fungicidal activity of *D. innoxia* to its content of tropane alkaloids [40], which are soluble in water. The bioactivity of *W. aristata* increased with the polarity of the solvent used, with the aqueous fraction being the most active against *B. cinerea*, exhibiting a slightly higher degree of mycelial inhibition (84.30%) compared to that of the ethanolic extract (72.22%). This fact could be attributed to the potential antifungal activity of some withanolide derivatives that are soluble in water. Thus, Seepe et al. [66] reported that a withaferin A glycoside isolated from a *Whitania* species was active against *Fusarium verticilloides.*

In general, except for *A. thuscula*, the liquid–liquid partition with solvents of different polarity increased the bioactivity against phytopathogens. Thus, the organic fractions (F1 and F2) from *C. symphytifolius* and *S. canariensis* significantly increased the potency against *B. cinerea,* whereas the bioactivity of the aqueous fractions from *D. innoxia* and *W. aristata* against this phytopathogen also increased, suggesting *B. cinerea* was sensitive to compounds of a different polarity and nature. The hexane and ethyl acetate fractions from *C. symphytifolius* and *S. canariensis* also exhibited a higher inhibition of *A. alternata’s* mycelial growth in comparison with the ethanolic extract.

## 3. Materials and Methods

### 3.1. Chemical and Reagents

All solvents used were analytical grade (Panreac, Barcelona, Spain). PGA culture medium (Potato glucose agar, Sigma Aldrich, St. Louis, MO, USA) and 9 cm diameter Petri dishes from Sarstedt (Nümbrecht, Germany) were used for the maintenance of fungal colonies and performance of the bioassays. Tetracycline (50 mg/L, Sigma Aldrich, St. Louis, MO, USA) was added to avoid bacterial growth on maintenance Petri dishes. Stock solutions of plant extracts were prepared with absolute ethanol (Sigma Aldrich, St. Louis, MO, USA).

### 3.2. Plant Material

Plant species tested in this study were collected in Tenerife and Gran Canaria (Canary Islands, Spain). The details of their collection, including the ecological state, place of collection, geographic coordinates, voucher specimens, and traditional uses of the sixteen plant species are summarized in Table 1 and Table 2. A voucher specimen of each species was deposited in the Herbarium of the University of La Laguna (TFG-SEGAI), and identified by Cristina González Montelongo, PhD. The plant material (leaves or aerial parts) was spread on a tray, turned over occasionally, and left to air-dry at room temperature (25 ± 4 °C) for two weeks. The dried plant materials were ground and stored in sealed bags to preserve them from moisture until extraction. 

### 3.3. Fungal Culture

Phytopathogenic fungi for bioassays belong to three cosmopolitan genera than cause serious damage to crops: *Alternaria alternata*, *Botrytis cinerea,* and *Fusarium oxysporum*. These fungi were maintained at 25 °C in darkness, and periodically replicated in Petri dishes with PGA culture medium and tetracycline to avoid the proliferation of contaminations. Strains of *B. cinerea* (B05.10) and *A. alternata* (Aa 100) were isolated from *Vitis vinifera* and *Lycopersicon esculentum*, respectively, both supplied by the Universidad de La Laguna, Tenerife. *F. oxysporum f. sp. lycopersici* (2715) strain, isolated from *Lycopersicon esculentum*, was provided by the Colección Española de Cultivos Tipo (CECT) from Valencia, Spain.

### 3.4. Plant Extracts Preparation

The air-dried, powdered plant materials (100 g) were extracted by repeated maceration (three times) with 1000 mL of ethanol (10 mL of ethanol/g of plant material) at room temperature for 24 h for each maceration process. The extract was filtered, and the solvent was removed under reduced pressure at 40 °C on a rotary evaporator, yielding green residue extracts. An aliquot (around 30 mg) of each extract was used for antifungal activity evaluation. 

### 3.5. Liquid–Liquid Partition Procedure of Selected Plant Extracts

The ethanolic plant extracts that showed the highest fungicidal inhibition (% I > 30%) in at least one of the tested fungi were selected for further fractionation with different solvents. Thus, they were sequentially fractionated by liquid–liquid partition using a series of solvents of increasing polarity by a modification of the Kupchan method [67]. Briefly, 5 g of solid ethanolic extract was suspended in 60 mL of distilled water and successively extracted with 60 mL of hexanes (3 × 60 mL) and ethyl acetate (3 × 60 mL). The organic phases were dried with Na_2_SO_4_, filtered, and concentrated under reduced pressure to afford the corresponding organic fractions. The aqueous phase was lyophilized, thereby yielding the aqueous fraction. All fractions from the liquid–liquid procedure were also assayed for their antifungal activity.

### 3.6. In Vitro Test-Assay on Mycelium

Antifungal activity was analyzed as mycelial growth inhibition by an agar-dilution method [68]. Once the culture medium solidified, eight 4 mm diameter discs of the target fungus per dish were placed in test and control Petri dishes. One dish per pathogen for each extract with eight sub-replicates in each dish was used. A stock solution of 50 mg/mL was prepared, using ethanol as solvent, and plant extracts were assayed at the concentration of 1 mg/mL. Colonies grown on Petri dishes were incubated for 48 h for *B. cinerea* and 72 h for *A. alternata* and *F. oxysporum* and were digitalized and measured with the application ImageJ (Image J, http://rsb.info.nih.gov/ij, accessed on 5 September 2022). Percent inhibition (% I) was calculated as % I = (C − T/C) × 100, where C is the diameter of the control colonies and T is the diameter of the test colonies. Those extracts showing a % I higher than 20% at a concentration of 1 mg/mL were then tested at lower doses (0.5 and 0.1 mg/mL). Methylparaben [57] was used as a positive control at the concentration of 1 mg/mL, whereas ethanol was used as a negative control, using one dish per pathogen and 8 discs for each control.

## 4. Conclusions

In summary, we identified five plant extracts with potential for application as biofungicides. Thus, the evaluation of the sixteen plant extracts’ activity against phytopathogenic fungi that cause serious damage to crops—*Alternaria alternata*, *Botrytis cinerea,* and *Fusarium oxysporum*—revealed that the ethanolic extracts from *S. canariensis* and *C. symphytifolius* were effective against the three phytopathogens and, therefore, could be promising broad-spectrum biofungicides. Moreover, the *A. thuscula*, *R. chalepensis,* and *W. aristata* extracts exhibited fungal specificity. These results show that a simple, easy, and reproducible procedure gave rise to extracts with a high potency towards inhibiting the mycelial growth of these phytopathogens. Therefore, this extraction procedure could be feasible for use by farmers against fungal infections. Overall, the liquid–liquid partition led to an improvement in the bioactivity of the fractions against the phytopathogens. Thus, the organic fractions (hexane and ethyl acetate fractions) from *S. canariensis* exhibited an % I greater than the ethanolic extract for all the tested fungi. Likewise, the growth inhibition of the organic fractions from *C. symphytifolius* was higher than its ethanolic extract against *A. alternata* and *B. cinerea*. Although none of the plant extracts (or their fractions) exhibited a higher growth inhibition than the control (methylparaben), these biopesticides are expected to be less persistent and harmful than synthetic ones, making their use more desirable.

These findings support the notion that plant-derived extracts are promising sources of biopesticides to applied in integrated pest management. The compounds that could contribute to the bioactivity of plant extracts have been suggested based on previous studies. Nevertheless, further studies will be conducted to identify these bioactive components and to investigate their potential application as biofungicides. 

## Figures and Tables

**Figure 1 plants-11-02988-f001:**
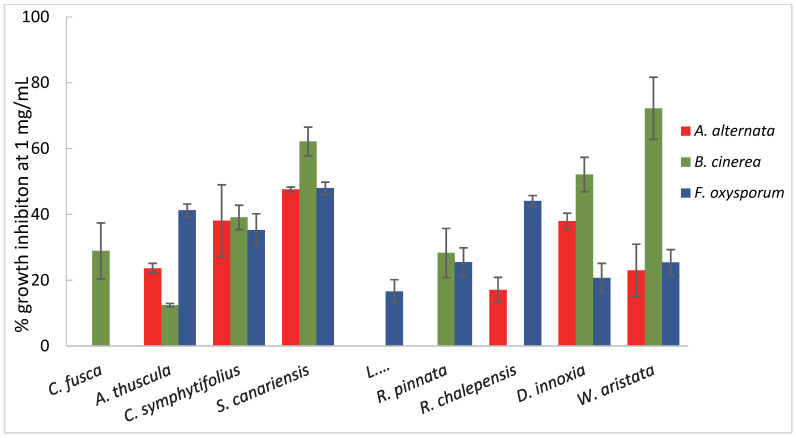
Antifungal effects (% growth inhibition) of plant extracts against *Alternaria alternata*, *Botrytis cinerea*, and *Fusarium oxysporum*. Results are expressed as percentage relative to the negative control. Methylparaben was used as a positive control (100% inhibition at 1 mg/mL). Data are presented as meas ± SD (Standard Deviation, n = 8).

**Figure 2 plants-11-02988-f002:**
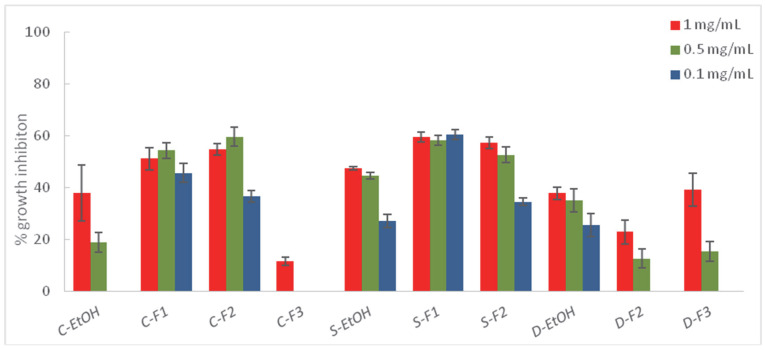
Antifungal effects (% growth inhibition) of the ethanolic extracts (EtOH) and fractions (F1: hexane fraction; F2: ethyl acetate fraction; F3: water fraction) from *C. symphytifolius* (C), *S. canariensis* (S), and *D. innoxia* (D) against *Alternaria alternata.* Results are expressed as percentage relative to the negative control. Methylparaben was used as a positive control (100% inhibition at 1 mg/mL). Data are presented as means ± SD (Standard Deviation, n = 8).

**Figure 3 plants-11-02988-f003:**
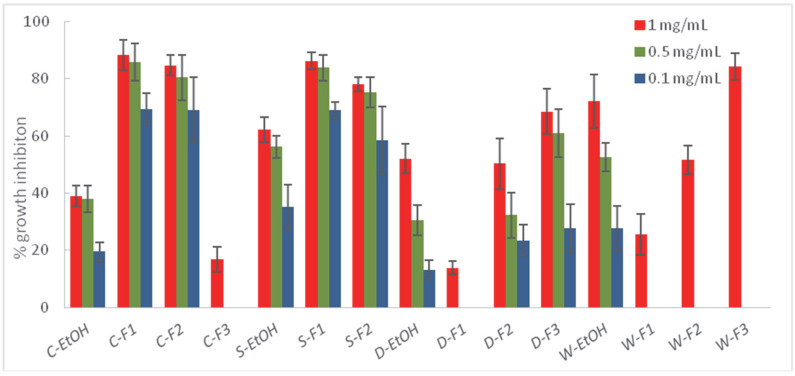
Antifungal effects (% growth inhibition) of the ethanolic extracts (EtOH) and fractions (F1: hexane fraction; F2: ethyl acetate fraction; F3: water fraction) from *C. symphytifolius* (C), *S. canariensis* (S), *D. innoxia* (D), and *W. aristata* (W) against *Botrytis cinerea.* Results are expressed as percentage relative to the negative control. Methylparaben was used as a positive control (100% inhibition at 1 mg/mL). Data are presented as means ± SD (Standard Deviation, n = 8).

**Figure 4 plants-11-02988-f004:**
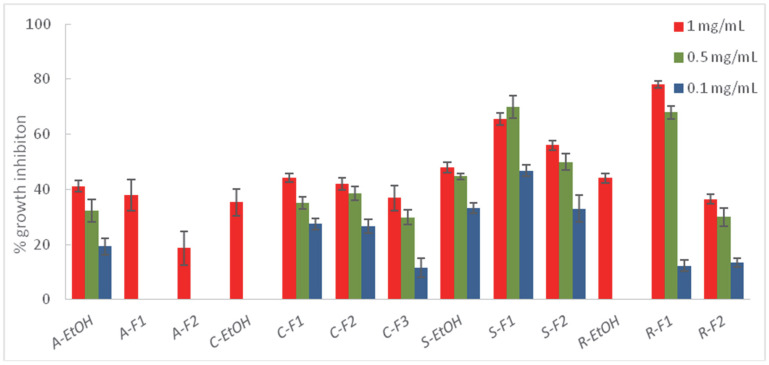
Antifungal effects (% growth inhibition) of the ethanolic extracts (EtOH) and fractions (F1: hexane fraction; F2: ethyl acetate fraction; F3: water fraction) from *A. thuscula* (A), *C. symphytifolius* (C), *S. canariensis* (S), and *R.chalepensis* (R) against *Fusarium oxysporum.* Results are expressed as percentage relative to the negative control. Methylparaben was used as a positive control (100% inhibition at 1 mg/mL). Data are presented as means ± SD (Standard Deviation, n = 8).

**Table 1 plants-11-02988-t001:** Details of collected plant species under study.

Family	Plant Species/Common Name/ Voucher Specimen ^1^	Ecological State ^2^/ Plant Organ	Collection Place ^3^/Geographic Coordinates ^4^/Date
Apocynaceae	*Ceropegia fusca* Bolle/ cardoncillo, TFC 53.261	Endemic/ Aerial part	Montaña Pelada, Granadilla de Abona/ 28°04′03″ N, 16°31′18″ W, 100 m.a.s.l./09-2018
Asteraceae	*Argyranthemum broussonetii* Pers. & Humphries/ magarza de monte/TFC 53.398	Endemic/ Aerial part	Ctra. El Bailadero, La Cumbrilla, Anaga/ 28°32′12″ N, 16°14′04″ W, 830 m.a.s.l./ 06-2020
	*Artemisia thuscula* Cav./ incienso/TFC 53.702	Endemic/ Aerial part	Ctra. Playa Santo Domingo, La Guancha/ 28°23′32″ N, 16°40′34″ W, 88 m.a.s.l./02-2021
Celastraceae	*Gymnosporia cassinoides* (L’Hér.) Masf. (=*Maytenus canariensis* (Loes.) G. Kunkel & Sunding)/peralillo/TFC 53.244	Endemic/ Leaves	Centro Ambiental La Tahonilla, La Laguna/ 28°28′34″ N, 16°19′16″ W, 574 m.a.s.l./ 09-2018
Cistaceae	*Cistus symphytifolius* Lam./ jara/TFC 53.703	Endemic/ Leaves	Pista Forestal El Lagar, La Guancha/ 28°20′42″ N, 16°39′15″ W, 1023 m.a.s.l./02-2019
Lamiaceae	*Lavandula canariensis* Mill./ lavanda, mato risco/TFC 53.400	Endemic/ Aerial part	Rambla de Castro, Los Realejos/ 28°23′40″ N, 16°35′32″ W, 125 m.a.s.l./05-2020
	*Salvia canariensis* L./ salvia, salvia morisca/TFC 53.328	Endemic/ Leaves	Gáldar/ 28°02′10″ N, 15°37′30″ W, 1415 m.a.s.l./06-2019
Lauraceae	*Apollonias barbujana* (Cav.) A. Braun subsp. *barbujana*/ barbusano, ébano de Canarias/ TFC 53.397	Endemic/ Leaves	La Laguna/ 28°28′48″ N, 16°19′11″ W, 560 m.a.s.l./ 01-2020
	*Laurus novocanariensis* Rivas- Mart. et al./loro, laurel/ TFC 53.399	Endemic/ Leaves	Cruz del Carmen, Anaga/ 28°31′63″ N, 16°16′48″ W, 970 m.a.s.l./ 06-2020
Rutaceae	*Ruta pinnata* L. f./ ruda salvaje/TFC 53.243	Endemic/ Leaves	La Laguna/ 28°28′44″ N, 16°19′21″ W, 575 m.a.s.l./09-2018
	*Ruta chalepensis* L./ ruda/TFC 53.756	Native to Macaronesian/Leaves	Araya, Candelaria/ 28°21′24″ N, 16°23′27″ W, 320 m.a.s.l./09-2018
Solanaceae	*Datura innoxia* Mill./ burladora, métel/TFC 53.216	Non-native in the Canary Islands/Leaves	La Orotava/ 28°23′40″ N, 16°32′46″ W, 198 m.a.s.l./ 03-2018
	*Datura stramonium* L./ estramonio, hierba del diablo, santas noches/TFC 53.217	Non-native in the Canary Islands/Leaves	La Orotava/ 28°23′40″ N, 16°32′46″ W, 198 m.a.s.l./ 03-2018
	*Nicotiana glauca* Graham/ bobo, tabaco moro/TFC 53.214	Invasive in the Canary Islands/Leaves	La Orotava/ 28°23′40″ N, 16°32′46″ W, 198 m.a.s.l./03-2018
	*Salpichroa origanifolia* (Lam.) Baill./ huevito de gallo/TFC 53.215	Non-native in the Canary Islands/Leaves	La Orotava/ 28°23′40″ N, 16°32′46″ W, 198 m.a.s.l./03-2018
	*Withania aristata* (Aiton) Pauquy/ orobal/TFC 53.219	Native to the Canary Islands/Leaves	Ctra. Playa Santo Domingo, La Guancha/ 28°23′33″ N, 16°40′32″ W, 88 m.a.s.l./04-2018

^1^ Voucher specimens were deposited at the Herbarium TFC (SEGAI) Universidad de La Laguna, Tenerife, Spain. ^2^ POWO (2021). Plants of the World Online. Royal Botanic Gardens, Kew. http://www.plantsoftheworldonline.org/, accessed on 9 July 2021. ^3^ All species were collected in Tenerife, except for *Salvia canariensis* collected in Gran Canaria, Canary Islands. ^4^ Geographic coordinates: latitude, longitude, and elevation.

**Table 2 plants-11-02988-t002:** Traditional uses and scientific reports of the plant species under study.

Plant Species	Traditional/Scientific Uses	References
*C. fusca*	Vulnerary, anti-hemorrhagic	[44]
*A. broussonetii*	Stomach remedy, anti-asthmatic, used to regulate menstrual disorders	[45]
*A. thuscula*	Insecticide, antiseptic, Protection of potato crops from insects	[46] [26]
*G. cassinoides*	Antiviral, antifungal Insecticidal, antifeedant against *Spodoptera littoralis*	[46] [27]
*C. symphytifolius*	Analgesic	[47]
*L. canariensis*	Vermifuge, antiparasitic, antiseptic Insecticide, anti-inflammatory, diuretic, sedative	[46] [26]
*S. canariensis*	Antiseptic, antiviral Antifungal, antibacterial	[44] [26]
*A. barbujana*	Diuretic, analgesic, antiulcerogenic, cardiotonic, expectorant, stomachic, sedative	[48]
*L. novocanariensis*	To treat bronchitis, pharyngitis, rheumatism Insecticidal	[26] [46]
*R. pinnata*	Antiparasitic, antiseptic Antibiotic, antifungal, anti-inflammatory, toxic	[46] [26]
*R. chalepensis*	Toxic Insecticide against *Sitophilus oryzae* Phytotoxic and antifungal activities Antibacterial and antifungal activity	[44] [49] [36,50] [51]
*D. innoxia*	Anti-asthmatic, anesthetic, toxic	[44]
*D. stramonium*	Antiasthmatic, antiparkinsonian, narcotic, antispasmodic, toxic Larvicidal and repellent against mosquitoes Anti-insect activity in Mediterranean plants Insecticide against *Tribolium castaneum*	[44,46] [52] [53] [54]
*N. glauca*	Dermatic, toxic Active against phytopathogenic fungi	[44] [41]
*S. origanifolia*	Anti-inflammatory, heal skin abrasions, diuretic, narcotic Isolated withanolides act as feeding inhibitors on *Tribolium castaneum*	[55] [42]
*W. aristata*	Diuretic, analgesic, anesthetic, anti-asthmatic, antirheumatic, antitumor, aphrodisiac Apocarotenoids and carotenoids showed phytotoxicity towards *Lepidium sativum*, *Lactuca sativa*, *Lycopersicum esculentum,* and *Allium cepa*	[26,46] [56]

## Data Availability

Not applicable.

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
