# Peer review of "Antifungal Potential of Canarian Plant Extracts against High-Risk Phytopathogens"

_plants, 2022, doi:10.3390/plants11212988_

Round 1
Reviewer 1 Report
The purpose of the reviewed paper was to study the antifungal activity of sixteen plant species from the Canary Islands Archipelago against the phytophatogenic fungi Botrytis cinerea, Fusarium oxysporum and Alternaria alternata.
Overall, this paper is well written and the experiments were well done. However, please make the following corrections before publishing:
1. article number 1 is about insecticides. I ask the Authors to identify publications that describe the harmfulness of fungicide active ingredients.
2. The literature list is extensive (lines 391-582). Since this is a research paper and not a review paper, I ask the Authors to cite articles related to the topic of the paper. I think it is necessary to remove from the references the works that do not deal with the effect of plant extracts on phytopathogens, that is, works number: 1, 31, 32, 39, 40, 42, 43, 44, 45, 46, 53, 54, 55, 56, 57, 58, 62, 63, 64, 69. Therefore, I ask you to correct the text concerning the listed works (lines 32-148) and Table 2.
3. lines 205-206 and 224-225: why the Authors did not write that the extract from R. chalepensis was also active against A. alternata, and the extract from L. novocanariensis was active against F. oxysporum?
4. could the Authors analyze which fungus was most sensitive to the plant extracts?
5. lines 337-346: please complete the description of the methodology, as it is unclear - how many dishes of mycelial discs were present (how many replicates), whether there were 8 discs per dish or 8 dishes of 1 disc each; what was the control colony; at what concentration was methylparaben used; were there relative control objects with ethanol? There are descriptions next to the figures, but they should also be given exactly in the methodology.
Author Response
Remark1. Article number 1 is about insecticides. I ask the Authors to identify publications that describe the harmfulness of fungicide active ingredients.
ANSWER: This point has been corrected in the revised version following the reviewer’s suggestion, and the reference 1 has been replaced by one that better fits the topic of the manuscript:
1. Zubrod, J.P.; Bundschuh, M.; Arts, G.; Bruhl, C.A.; Imfeld, G.; Knabel, A.; Payraudeau, S.; Rasmussen, J.J.; Rohr, J.; Scharmuller, A.; Smalling, K., Stehle, S.; Schulz, R.; Schäfer, R.B. Fungicides: An overlooked pesticide class? Environ. Sci. Technol. 2019, 53, 3347–3365, doi: 10.1021/acs.est.8b04392.
Remark 2. The literature list is extensive (lines 391-582). Since this is a research paper and not a review paper, I ask the Authors to cite articles related to the topic of the paper. I think it is necessary to remove from the references the works that do not deal with the effect of plant extracts on phytopathogens, that is, works number: 1, 31, 32, 39, 40, 42, 43, 44, 45, 46, 53, 54, 55, 56, 57, 58, 62, 63, 64, 69. Therefore, I ask you to correct the text concerning the listed works (lines 32-148) and Table 2.
ANSWER: This point has been corrected in the revised version following the reviewer’s suggestion. Thus, the indicated references have been removed and the rest of references renumbered.
Remark 3. Lines 205-206 and 224-225: why the Authors did not write that the extract from R. chalepensis was also active against A. alternata, and the extract from L. novocanariensis was active against F. oxysporum?
ANSWER: The reviewer is right, and this point has been corrected in the revised manuscript:
In lines 180: … were, C. fusca, A. thuscula, C. symphytifolus, S. canariensis, L. novocanariensis, R. pinnata, R. chalepensis, D. innoxia and W. aristata.
In lines 207-209: L. novocanariensis showed a low activity against F. oxysporum (% I of 16.69 at 1 mg/mL), whereas was inactive against the other tested fungi (% I < 10 mg/mL).
In lines 211-213: In the case of the other Rutaceae species studied, R. chalepensis showed a high activity against F. oxysporum (% I of 44.09% at 1 mg/mL), and a low one on A. alternata (% I of 17.08 at 1 mg/mL).
Remark 4. Could the Authors analyze which fungus was most sensitive to the plant extracts?
ANSWER: Regarding this point, a paragraph has been included at the end of the Section 2.2. (pg 7, lines 234-240) in the revised manuscript:
“Furthermore, the most sensitive fungus to the plant extracts seems to be B. cinerea, following by F. oxysporum, whereas A. alternata turned out to be the least sensitive to the studied plants. Thus, seven plant extracts showed a growth inhibition higher than 20% on F. oxysporum compared to six on B. cinerea. However, % I values against B. cinerea were higher (28.91-72.22% at 1 mg/mL). Moreover, five plant extracts showed a % I > 20% against A. alternata with values ranging from 23.63 to 47.60 % at 1 mg/mL.”
Remark 5. Lines 337-346: please complete the description of the methodology, as it is unclear - how many dishes of mycelial discs were present (how many replicates), whether there were 8 discs per dish or 8 dishes of 1 disc each; what was the control colony; at what concentration was methylparaben used; were there relative control objects with ethanol? There are descriptions next to the figures, but they should also be given exactly in the methodology.
ANSWER: This point has been corrected in the revised version, and the Section 3.6. In vitro Test-assay on Mycelium was completed as follows, and a reference [68]for the methodology used has been included:
“Antifungal activity was analyzed as mycelial growth inhibition by an agar dilution method [68]. Once the culture medium solidified, eight 4 mm diameter discs of the target fungus per dish were placed, in test and control Petri dishes. A dish per pathogen for each extract with eight subreplicates in each dish was used. The stock solution of 50 mg/mL were prepared, using ethanol as solvent, and plant extracts were assayed at the concentration of 1 mg/mL. Colonies grown on Petri dishes incubated for 48 h for B. cinerea and 72 h for A. alternata and F. oxysporum were digitalized and measured with the application ImageJ (Image J, http://rsb.info.nih.gov/ij). Percent inhibition (% I) was calculated as: % I = (C - T/C) × 100, where C is the diameter of the control colonies and T is the diameter of the test colonies. That extracts showing a % I higher than 20% at a concentration of 1 mg/mL were then tested at lower doses (0.5 and 0.1 mg/mL). Methylparaben [57] was used as a positive control at the concentration of 1 mg/mL, whereas ethanol was used as a negative control, using a dish per pathogen and 8 discs for each control.”
68. Cosoveanu, A.; Hernandez, M.; Iacomi-Vasilescu, B.; Zhang, X.; Shu, S.; Wang, M.; Cabrera, R. Fungi as endophytes in Chinese Artemisia spp.: juxtaposed elements of phylogeny, diversity and bioactivity. Mycosphere 2016, 7, 102-117, doi: 10.5943/mycosphere/7/2/2 .
Reviewer 2 Report
The review manuscript with the title "Antifungal Potential of Canarian Plant Extracts against High-risk Phytopathogens in Crops" addresses an interesting, current topic.
The abstract presents an overview of the entire MS.
The introduction presents relevant information from specialized literature.
The results are well organized and structured. These are analyzed statistically. The statistical analysis seems to be appropriate.
The methods are sufficiently detailed.
The conclusion could be improvement according to the presented results.
Minor comments:
1. I recommend consulting a bibliographic source:
Romina Alina Marc, Implications of Mycotoxins in Food Safety, DOI: 10.5772/intechopen.102495
2. I recommend adding in the introduction the general information about the mycotoxins described in the manuscript.
3. I recommend the detailed description of the conclusions according to the results obtained.
Author Response
Remark 1. I recommend consulting a bibliographic source: Romina Alina Marc, Implications of Mycotoxins in Food Safety, DOI: 10.5772/intechopen.102495
ANSWER: The suggested reference has been included in the revised manuscript (lines 46-47, reference 9), together with a sentence regarding mycotoxins. “Moreover, mycotoxin contamination decreases product quality and has repercussions on human health and significant economic losses [9].”
9. Marc, R.A. Implications of Mycotoxins in Food Safety. In Mycotoxin and Food
Safety-Recent Advances.; Marc, R.A., Ed.; IntechOpen, 2022; pp. 1-32, ISBN 978-1- 83962-904-4, doi: 10.5772/intechopen.102495.
Remark 2. I recommend adding in the introduction the general information about the mycotoxins described in the manuscript.
ANSWER: Our work deals with the potential of plant extracts on phytopathogenic fungi, activity that is expected to be linked to the plant components (metabolites). Theretofore, the manuscript does not deal with mycotoxins, and this topic is out to the present work. Even though, a paragraph regarding the main mycotoxins produced by the phytopathogens tested in this study, and responsible for their phytotoxicity has been added, in addition to the corresponding references.
In lines: 51-52: “The sequisterpenes botrydial and botcinin acid are the two major
phytotoxins involves in the infection of this fungus [11].”
11. Dalmais, B.; Schumacher, J.; Moraga, J.; Le Pêcheur, P.; Tudzynski, B.; Collado, I.G.; Viaud, M. The Botrytis cinerea phytotoxin botcinic acid requires two polyketide synthases for production and has a redundant role in virulence with
botrydial. Mol. Plant Pathol. 2011, 12, 564–579, doi:10.1111/j.1364- 3703.2010.00692.x.
In lines 58-59: …produce a range of mycotoxins, including enniatins, fusaric acid and moniliformin [13].
13. Munkvold, G.P. Fusarium Species and Their Associated Mycotoxins. In Mycotoxigenic Fungi: Methods and Protocols; Moretti, A., Susca, A., Eds.; Springer Science-Business Media LLC: New York, NY, USA, 2017; pp. 51–106 ISBN 978-1-4939-6705-6.
In lines 63-64: The main mycotoxins found in rotten samples of crops infected by this fungus are altenuene, alternariol, and altertoxins [15].
15. Fernández Pinto, V.E.; Patriarca, A. Alternaria species and their associated mycotoxins. In Mycotoxigenic Fungi: Methods and Protocols; Moretti, A., Susca, A., Eds.; Springer Science-Business Media LLC: New York, NY, USA, 2017; pp. 13–32 ISBN 978-1-4939-6705-6.
Remark 3. I recommend the detailed description of the conclusions according to the results obtained.
ANSWER: This point has been corrected in the revised version, including a paragraph that clarifies the conclusions (lines 376-379).
“Thus, the organic fractions (hexanes and ethyl acetate fractions) from S. canariensis exhibited %I greater than the ethanolic extract for all the tested fungi. Likewise, the growth inhibition of the organic fractions from C. symphytifolius was higher than its ethanolic extract against A. alternata and B. cinerea.
Reviewer 3 Report
Reviewer report for paper entitled "Antifungal Potential of Canarian Plant Extracts against High-risk Phytopathogens in Crops". This study was focused on the assessment the potential uses of some Canarian Plant extract as potential agent for fungal plant disease control based on their antifungal properties. This type of study is of interest as its online with the increased trend to replace the toxic antifungal compounds with safe natural compounds. This study of potential interest for both scientific society and plantation when further studied in field application.
General comment: The main missing part of this study is the field study and application of the isolated fractions in (in vivo) using plant study. Also, the isolation of the specific bioactive fractions from these plant to understand in depth how it work as antifungal. However, this study can be considered as the initial stage for large platform technology for development of plant based antifungal compounds for agriculture application.
Minor Comments:
- Title: I recommend to remove (in crops) from the title as there were no field study undertaken.
- Introduction part is well written and provided the necessary information as introduction for this type of research.
- Materials and Methods is acceptable. However, need to provide full information about the chemical used (Grade, Company, City, Country) as well as for the equipment used (Model, Company, City, Country). Authors need to provide a reference for all methods used in this study.
- Results and discussion: Based on Tables 1, authors screened 16 different plant for their potential antifungal activities. This table can be used in materials and methods part not in results part as it shows the source of biomaterials used. In addition, line 106-148 can not be in results part its just reviewing of the previous published work about these plant and no results presented at all. In addition, the data in table 1 is not in line with the results of Figure 1 which shows only the results of 9 plants!. In addition, table 2, (potential medicinal uses of the selected plants) can not be in results and discussion part. The data in this table can be used in part in the discussion part.
- In addition for this type of study, photos of fungal inhibition is needed to be presented (either in text or in supplementary files).
- In depth discussion of the data for they why? it inhibit fungal growth, what type of potential compounds in the solvent fraction is totally missing. Therefore, in depth discussion is missing in this research.
- Therefore, this work needs careful major revision before considering for publication
Author Response
Remark 1. General comment: The main missing part of this study is the field study and application of the isolated fractions in (in vivo) using plant study. Also, the isolation of the specific bioactive fractions from these plants to understand in depth how it work as antifungal. However, this study can be considered as the initial stage for large platform technology for development of plant based antifungal compounds for agriculture application.
ANSWER: I agree with the reviewer’s comment, and future works will be focused on the potential application of selected plant extracts in field. Therefore, this point will be the aim of a future work.
Minor Comments:
Remark 2. - Title: I recommend to remove (in crops) from the title, as there was no field study undertaken.
ANSWER: This point has been corrected in the revised version following the reviewer’s suggestion.
Remark 3. - Materials and Methods is acceptable. However, need to provide full information about the chemical used (Grade, Company, City, Country) as well as for the equipment used (Model, Company, City, Country). Authors need to provide a reference for all methods used in this study.
ANSWER: This point has been corrected in the revised version following the reviewer’s suggestion. A reference for the liquid-liquid partition procedure [67] is already in the current version of the manuscript, and a reference for the methodology used the in vitro assay on mycelium has been included in the revised manuscript [68].
Remark 4. - Results and discussion: Based on Tables 1, authors screened 16 different plant for their potential antifungal activities. This table can be used in materials and methods part not in results part as it shows the source of biomaterials used. In addition, line 106-148 can not be in results part its just reviewing of the previous published work about these plant and no results presented at all. In addition, the data in table 1 is not in line with the results of Figure 1 which shows only the results of 9 plants. In addition, table 2, (potential medicinal uses of the selected plants) can not be in results and discussion part. The data in this table can be used in part in the discussion part.
ANSWER: The selection of the species aim of this work is based on the traditional uses (ethnobotany) and previous reports about their potential as biopesticides. Therefore, the authors consider that the information provided in Tables 1 and 2 are essential, since justified and are the basis of this work. The details of plants collection, which include the collection data and the part of the plant to study, are linked to the obtained results. In addition, Table 2 gives the necessary arguments for the selection of plants to study, and a Discussion on these species and the reported metabolites that could be responsible for their activity is included in the text. Thus, the information gives in the tables allows us the analysis and discussion of the results. Therefore, the authors consider that discussion on selected plants is more appropriate in the Results and Discussion Section than in the Materials and Methods Section. Regarding Figure 1, only the results of the 9 plants that showed some degree of activity (% inhibition growth > 10%) were included. However, information of the evaluation for the 16 plant extracts is included in the Supporting Information (Table S1. Antifungal effects (% growth inhibition) of plant extracts against Alternaria alternata, Botrytis cinerea and Fusarium oxysporum). Moreover, Table S2. Antifungal effects (% growth inhibition) of selected plant fractions against Alternaria alternata, Botrytis cinerea and Fusarium oxysporum, corresponding to the liquid-liquid partition of most active plant extracts. The authors think that graphics (Figures 1-3) representing the most relevant results are more appropriate in the text of the manuscript that the corresponding tables.
Remark 5. - In addition for this type of study, photos of fungal inhibition is needed to be presented (either in text or in supplementary files).
ANSWER: This point has been corrected in the revised version following the reviewer’s suggestion, and photos of fungal inhibition assays have been included in the Supporting Information (Figure S1).
Remark 6. - In depth discussion of the data for they why? it inhibit fungal growth, what type of potential compounds in the solvent fraction is totally missing. Therefore, in depth discussion is missing in this research.
ANSWER: In the results and discussion there are several comments regarding the type of compounds that have been reported in the study plants.
Lines 91-94: “A large amount of phytochemicals reported from medicinal plants, such as alkaloids, phenylpropanoids, terpenoids, flavonoids or saponins, have demonstrated to be effective in crop protection, and therefore, the study of medicinal plant extracts [24] is becoming a field of growing interest as part of an integrated pest management.
Lines 114-119: Several compounds (sesquisterpenes) isolated from Gymnosporia cassinoides, formerly known as Maytenus canariensis, have shown both antifeedant and insecticidal activities against larvae of Spodoptera littoralis [27,28].
Lines 125-126: Nevertheless, the presence of compounds such as diterpenes [31] and triterpenes [32] with known fungicidal properties makes their study of interest. Lines 130-131: Essential oils of R. chalepensis have showed fungicide activity against fungi of Fusarium [35], Alternaria [36] and Aspergillus [37] genus.
Line 137-138: S. origanifolia is characterized by having a high content of
withanolides...
Lines 139-140: Species of the genus Withania, which includes W. aristata, are also a rich source of withanolides [43]. The high content of this type of metabolites in both S. origanifolia and W. aristata suggest these plants could be promising candidates as source of natural fungicides against the selected phytopathogenic species.” Moreover, in the conclusion is statement: “The compounds that could contribute to the bioactivity of plant extracts are suggested on the basis of previous studies. Nevertheless, further studies will be conducted to identify these bioactive components and to investigate their potential application as biofungicides.”
Therefore, we consider that this point is widely discussed in the current version of the manuscrip
Round 2
Reviewer 3 Report
Authors did significant change to improve the manuscript and answered all raised issues point by point. I recommend to accept this paper for publication.